# The Evaluation of 17 Gastrointestinal Tumor Markers Reveals Prognosis Value for MUC6, CK17, and CD10 in Gallbladder-Cancer Patients

**DOI:** 10.3390/diagnostics11020153

**Published:** 2021-01-21

**Authors:** Cristian Carrasco, Andrés Tittarelli, Natalia Paillaleve, Maeva Del Pozo, Daniel Rojas-Sepúlveda, Omar Barría, Paula Fluxá, Melissa Hott, Carolina Martin, Claudia Quezada, Flavio Salazar-Onfray

**Affiliations:** 1Subdepartamento de Anatomía Patológica, Hospital Base de Valdivia, Valdivia 5090000, Chile; napvergara@gmail.com (N.P.); maevadelpozol@msn.com (M.D.P.); 2Programa Institucional de Fomento a la Investigación, Desarrollo e Innovación, Universidad Tecnológica Metropolitana, Santiago 8940577, Chile; atittarelli@utem.cl; 3Millennium Institute on Immunology and Immunotherapy, Faculty of Medicine, Universidad de Chile, Santiago 8380453, Chile; daniel.rs@qf.uchile.cl (D.R.-S.); omar.barria@ug.uchile.cl (O.B.); claudiaquezada@uach.cl (C.Q.); 4Disciplinary Program of Immunology, Institute of Biomedical Sciences, Faculty of Medicine, Universidad de Chile, Santiago 8380453, Chile; 5Departamento de Cirugía Oriente, Facultad de Medicina, Universidad de Chile, Santiago 7500922, Chile; fluxapaula@gmail.com; 6Subdepartamento de Enfermedades Virales, Instituto de Salud Pública, Santiago 7780050, Chile; mhott@ispch.cl; 7Escuela de Tecnología Médica, Universidad Austral de Chile, Puerto Montt 5500000, Chile; karolit_81@hotmail.com; 8Instituto de Bioquímica y Microbiología, Universidad Austral de Chile, Valdivia 5090000, Chile

**Keywords:** gallbladder cancer, prognosis, biomarkers, mucin 6, cytokeratin 17, CD10, immunohistochemistry, tissue microarray

## Abstract

Gallbladder cancer (GBC) is an aggressive and highly lethal disease with relatively low global incidence, but one that constitutes a major health problem in Asian and Latin American countries, particularly in Chile. The identification of new tumor-associated markers with potential prognosis value is required for GBC clinical practice. Using immunohistochemistry/tumor tissue microarray, we evaluated the expression of 17 gastrointestinal tumor-associated protein markers (CK7, CK17, CK19, CK20, CKLMW, CKHMW, MUC1, MUC2, MUC5AC, MUC6, CA125, CD10, CEA, vimentin, villin, claudin-4, and CDX2) in primary gallbladder adenocarcinomas from 180 Chilean patients and analyzed potential associations with their pathological and clinical characteristics. Younger female patients with well- to moderately differentiated tumors had a better prognosis than that of older female or male patients with tumors with a similar tumor differentiation grade. Among all analyzed markers, MUC6 expression was associated with better prognosis in patients with well- to moderately differentiated tumors, whereas CK17 or CD10 was associated with worse prognosis in patients with poorly differentiated tumors. In addition, the MUC6^+^CK17^–^ expression pattern was strongly associated with better prognosis in patients with well- to moderately differentiated tumors, whereas patients with poorly differentiated tumors and with the CK17^+^CD10^+^ expression pattern showed worse prognosis. Our results suggest that tumor MUC6, CK17, and CD10 can be considered as potential prognosis markers for GBC.

## 1. Introduction

Gallbladder cancer (GBC) is the most frequent malignancy of the biliary tract and the fifth most common digestive neoplastic disease [1,2]. The prognosis of GBC is poor, with a median overall survival (OS) time of 3–11 months and a five-year survival rate ranging from 4% to 60%, depending on disease stage and tumor receptibility. This is mainly due to the aggressive behavior of the tumor cells and its nonspecific symptoms, leading to late clinical manifestation and, therefore, diagnosis at an advanced stage [3].

Global incidence and mortality for GBC show a strong geographical distribution and ethnic correlation [1,2,4]. In 2018, the International Agency for Research on Cancer (IARC) Globocan 2018 database estimated that GBC constitutes 1.7% of all cancer deaths, with 220,000 new cases diagnosed annually [2]. Interestingly, GBC is infrequent in most Western countries, while Japan, India, and Latin American countries share a high incidence of this tumor, constituting a major health problem [4,5,6]. Remarkably, Chile has the highest global incidence and mortality rates of GBC, representing one of the main causes of death due to malignant tumors in women [3,6]. The development of GBC was linked to various predisposing environmental and genetic variants [7]. However, one of the most relevant risk factors for GBC development is cholecystolithiasis and chronic inflammation of the gallbladder [8]. Other risk factors associated with GBC carcinogenesis are, for example, gender (women have two to six times more chances to develop GBC than men), obesity, congenital developmental abnormalities, sedentary behavior, gallbladder chronic infection and inflammation, ethnicity, and advanced age [7,8,9]. Adenocarcinoma is the most common histologic type, accounting for 98% of all gallbladder tumors.

Traditional chemotherapy and radiotherapy managements provide only marginal survival benefits to patients with locally advanced or metastatic GBC [10,11]. Therefore, the identification of novel molecular markers is crucial for the early diagnosis, prognosis, and further development of targeted therapies for GBC patients [12]. In this study, in a cohort of 180 Chilean patients with primary gallbladder adenocarcinoma, we evaluated the expression of 17 gastrointestinal tumor-associated protein markers consisting of six cytokeratins (CK7, CK17, CK19, CK20, CKLMW, and CKHMW), five mucins (MUC1, MUC2, MUC5AC, MUC6, and CA125), two glycoproteins (CD10 and CEA), two cytoskeleton-associated proteins (vimentin and villin), one tight junction protein (Claudin4), and one transcription factor (CDX2). The association of the expression profile of these markers with clinical and pathological GBC patients’ characteristics was further analyzed.

Cytokeratin expression is frequently used for the differential diagnosis of carcinomas originating from different sites, with the combination of CK7 and CK20 being the most useful for this purpose [13]. Mucins play a pivotal role in gallstone pathogenesis and in influencing cancer biology, such as carcinoma invasion and metastatic potential [14,15]. Glycoproteins CD10 and carcinoembryonic antigen (CEA) were used as important biomarkers for digestive-tract tumors [16,17]. Both vimentin and villin are proteins associated with GBC and involved in cytoskeleton homeostasis, epithelial-to-mesenchymal transition, and epithelial maintenance [18,19]. CDX2 is a transcription factor related to cellular growth and the differentiation of the intestinal tract, and it was previously associated with the clinical outcome of GBC patients [20].

Our results showed that MUC6 tumor expression was associated with a better prognosis in patients with well- to moderately differentiated tumors, whereas CK17 or CD10 tumor expression was associated with worse prognosis in patients with poorly differentiated tumors. Therefore, we propose that these proteins may constitute candidates for prognosis markers in patients with primary adenocarcinoma of the gallbladder.

## 2. Materials and Methods

### 2.1. Patients and Tumor Samples

Retrospective analysis of cholecystectomy specimens diagnosed as GBC and clinical data from patients diagnosed at the pathological anatomy subdepartment of Hospital Base Valdivia (Valdivia, Chile) was performed from 2001 to 2015. Cases were selected using code C.23 according to the International Classification of Diseases for Oncology (ICD-O). The study was performed in agreement with the Code of Ethics of the World Medical Association (Declaration of Helsinki), printed in the *British Medical Journal* (18 July 1964), and approved by the Bioethical Committee for Human Research of the Valdivia Regional Hospital. All patients signed a letter of informed consent for publication at the time of surgery.

Only primary invasive gallbladder adenocarcinoma cases were considered, reaching 215 cases. In situ adenocarcinoma, squamous carcinoma, neuroendocrine carcinoma, and metastases were excluded. From the 215 clinical cases, 180 samples (83.7%, slides and paraffin tissue blocks) were obtained. For the detection of each antigen, 159–178 cases could be analyzed due to limitations of the availability of tissue material. For each tumor sample, 2–3 foci of the neoplastic invasion were selected for analysis under light microscopy, and these areas were marked.

Tumor staging and differentiation status were based on the reported imaging findings and the pathologic evaluation of the resected specimens according to the eighth edition of the *American Joint Committee on Cancer Staging Manual* [21].

### 2.2. Tissue Microarray

Quick-Ray^®^ UT06 Manual Tissue Microarrayer (Unitma Co., Ltd., Seoul, Korea) was used according to the manufacturer’s instructions. We used a 2 mm puncher tip with its respective puncher and receptor blocks with perforations of 2 mm × 60 mm. When the slices were faced with their respective donor block (the tissue to be studied in a paraffin block), the chosen site could be precisely located. Previous to tissue extraction, donor blocks were incubated at 37 °C–40 °C for 15 min–20 min. Then, samples were introduced into a puncher tip at a depth of 5 mm (needle measurement), obtaining a tissue cylinder that was deposited into a receptor block.

The block was placed on a horizontal and flat surface, and the needle of the puncher tip was accommodated in a position perpendicular to the holes. Tissue was slowly injected into the cylinder of the corresponding hole. Lastly, cylinders were softly pushed to even out the superior surface of the receptor block. The receptor blocks for the microarrays had 60 cylinder perforations, of which only 59 were used each time, leaving a strategic place to individualize and orientate them in the direction in which they were mounted, facilitating their identification further. In parallel, a sheet with a grid was created where the position of each piece of tissue and its respective case number were recorded. The receptor block was placed on a base mold (from the kit) face down and incubated at 70 °C for 30 min–60 min. Once the blocks had been completely transparent, they were embedded in an inclusion cassette and left to solidify on a cold plaque. Finally, a rotary microtome was used to produce slices with a width of three microns. These slices were placed on slides of 75 mm × 25 mm with their respective positive tissue controls.

### 2.3. Immunohistochemistry

Representative blocks (tissue specimens fixed in 10% neutral buffered formalin and embedded in paraffin wax) were selected for immunohistochemistry analysis. Paraffin blocks were dissected into 3 μm thick slides mounted on positively charged slides. The technique was achieved with an automatic BenchMark GX Ventana system (Roche, AZ, USA) using an ultraView Universal DAB Detection kit (Roche, Arizona, USA), following the manufacturer’s instructions. A complete list of the antibodies and positive control tissue used in this work is shown in Appendix A. Negative controls were prepared by omitting primary antibodies. Additionally, negative tissue controls were routinely used to reveal nonspecific binding and false-positive results. Two independent pathologists blindly evaluated all immunohistochemistry slides under light microscopy and classified antigen expression by comparison with positive and tissue controls included in each slice. The intensity of antigen expression was determined with a comparison among the positive tissue controls included in each slice and subjectively graded as negative expression—weak expression for those cases where the staining intensity was lower than the positive control, moderate expression for those samples with an antigen staining intensity equal to the positive control, and intense expression for those cases when antigen-staining intensities were higher than the positive control. Moreover, positive staining in tissue was subjectively categorized as focal, patchy, or diffuse patterns according to the marker location. Intensity and pattern distributions of the evaluated markers in primary gallbladder adenocarcinoma are shown in Appendix A. For statistical analysis of associations between tumor-associated marker expression and the clinicopathological characteristics of GBC patients, only the absence or presence of positive staining was considered.

### 2.4. Statistical Analysis

Statistical analyses were performed using GraphPad Prism 7 (GraphPad Software Inc., San Diego, CA, USA) and STATA/IC 16.0 (StataCorp, College Station, TX, USA) software. Chi-squared with Yates correction analysis was used to analyze the frequency of each group in relation to the studied parameters. Kaplan–Meier and log-rank (Mantel–Cox) tests were used to construct and evaluate OS data. Univariate and multivariate analyses were performed using the Cox proportional-hazard regression model (Breslow method) to study the effects of different variables on OS. Differences were considered statistically significant at *p* < 0.05.

## 3. Results

### 3.1. Clinicopathological Characteristics of GBC Patients

In line with previously reported data [1,3], the majority of GBC patients were female (77.8%) (Table 1). The median age of patients was 67.5 years old, ranging from 34 years old to 92 years old. The median age of the female patients was 66.5 years old, while the male median age was 70 years old. According to tumor differentiation status, only 8.9% of patients had well-differentiated tumors, whereas 58.3% and 32.8% had moderately and poorly differentiated tumors, respectively (Table 1). Patients with well-differentiated tumors were, on average, younger than patients with moderately or poorly differentiated tumors (60 years old vs. 69 years old or 67 years old, respectively). Additionally, the frequency of patients with advanced-stage tumors (T2–T4) was higher in patients with poorly differentiated tumors (100%) than in patients with moderately (87%) or well-differentiated tumors (55%) (Table 1).

As expected, the median OS time and five-year survival rate significantly decreased as the tumors became less differentiated (Figure 1A). In fact, patients with poorly differentiated tumors showed a median OS of 5.5 months and a five-year survival rate of 8.6%, while patients with well- to moderately differentiated tumors had a median OS of 14.5 months and a five-year survival rate of 26.3% (*p* < 0.0001; Figure 1B). In addition, female patients with well- to moderately differentiated tumors showed significantly better prognosis than male patients with tumors with the same grade of differentiation did (*p* = 0.0354; median OS, 19.5 months vs. 12 months; five-year survival rate, 29.3% vs. 15.4%; Figure 1C). Gender was not associated with the median OS of patients when tumors progressed to being poorly differentiated, but the percentage of long-term survival patients (five-year survival rate) is superior in women (10.9%) to male patients (0%) with poorly differentiated tumors (Figure 1D).

Analysis based on age, disease onset, and gender showed that younger female patients (early onset) with well- to moderately differentiated tumors have a better prognosis than male (both early and late-onset) and late-onset female (Figure 1E,F) patients. On the other hand, the age of disease onset and gender did not significantly correlate with the survival of patients with poorly differentiated tumors (Figure 1G,H). Altogether, according to our GBC patient cohort, younger female patients with well- to moderately differentiated tumors have a better prognosis than that of older female or male patients.

### 3.2. Immunohistochemistry Analysis of Broad Panel of Tumor-Associated Markers in Primary GBC Samples

The distribution of staining patterns (positive cases) for the 17 tumor-associated immunohistochemistry markers according to tumor differentiation and staging is shown in Table 2. Our results showed that more than 90% of cases diagnosed as primary adenocarcinoma of the gallbladder were positive for CK7 (97.6%), CK19 (98.2%), CKLMW (100%), CKHMW (91.6%), or MUC1 (97.1%) expression, or were negative for vimentin (96.1%) or MUC2 (96.9%). Expression levels from the rest of the evaluated tumor-associated markers (CK17, CK20, MUC5AC, MUC6, CDX2, CEA, CA125, CD10, claudin-4, and villin) showed more variable behavior, ranging from 81.8% (MUC5AC) to 15.4% (CK20) of positivity. Vimentin expression was statistically correlated with tumor differentiation status; it was expressed in a higher proportion in poorly differentiated tumors compared with well- to moderately differentiated tumors (*p* = 0.0014). Additionally, CA125 was expressed in a higher proportion in advanced-stage (T2 to T4) than in early stage (tumor in situ (TIS)–T1) tumors (*p* = 0.009), whereas MUC6 was expressed in higher proportion in early stage tumors (*p* = 0.04) (Table 2).

### 3.3. Association of Tumor-Associated Markers with GBC Patient Prognosis

We analyzed the association between molecular markers’ expression patterns and median OS according to the tumor differentiation status and gender of GBC patients. For these analyses, we excluded tumor-associated markers expressed in more than 90% (CK7, CK19, CKLMW, CKHMW, and MUC1) or less than 10% (vimentin and MUC2) of the tumor samples. Examples of immunohistochemistry photographs of tumor samples positive for CK7, CK19, CKLMW, CKHMW, and MUC1 or negative for vimentin and MUC2 are shown in Appendix A. Expression frequencies for CK20, MUC5AC, CDX2, CEA, CA125, or claudin-4 were not statistically associated with the OS time of GBC patients (data not shown). Immunohistochemistry examples of tumor samples positive for CK20, MUC5AC, CDX2, CEA, CA125, and claudin-4 are shown in Appendix A.

MUC6 expression in well- to moderately differentiated tumors was significantly correlated with better prognosis in GBC patients (median OS, 26 month vs. 9 months; *p* = 0.032), particularly in women (median OS, 34 months vs. 9 months; *p* = 0.0035) (Figure 2A). However, MUC6 expression was not associated with survival in patients with poorly differentiated tumors, independent of their gender (Appendix A). In addition, CK17 expression in well- to moderately differentiated tumors was only correlated with worse prognosis in male patients (median OS, 9 months vs. 143 months; *p* = 0.04) (Appendix A), in GBC patients with poorly differentiated tumors (median OS, 4.5 months vs. 11 months; *p* = 0.0046), and particularly in women (median OS, 5 months vs. 13.5 months; *p* = 0.0049) (Figure 2B). The expression of CD10 was slightly associated with worse prognosis in GBC patients with poorly differentiated tumors (median OS, 3 months vs. 7 months; *p* = 0.0208), while its expression was not associated with differential prognosis in patients with well- to moderately differentiated tumors or with patient gender (Figure 2C and data not shown). Representative immunohistochemistry images of tumor samples positives for MUC6, CK17, and CD10 are shown in Figure 2D.

Univariate and multivariate analyses of patients with poorly differentiated tumors using Cox regression models showed that CK17 and CD10 tumor expressions might be independent predictors for OS on GBC patients (Table 3). On the other hand, the same analyses in patients with well- to moderately differentiated tumors showed that MUC6 tumor expression and age of disease onset could be independent predictors in GBC patients (Table 3).

Then, we analyzed the prognosis association of a combined tumor-marker expression panel. The MUC6^+^CK17^–^ expression pattern was strongly associated with better prognosis in GBC patients with well- to moderately differentiated tumors as compared with the expression of the MUC6^–^CK17^+^ pattern (median OS, 75 months vs. 9 months; *p* = 0.0037) (Figure 3A). Additionally, patients with poorly differentiated tumors expressing double marker patterns CK17^+^Villin^–^, CD10^+^Villin^–^, or CK17^+^CD10^+^ showed lower OS times than patients with opposite double-marker patterns CK17^–^Villin^+^ (3 months vs. 10 months; *p* = 0.0156), CD10^–^Villin^+^ (2.5 months vs. 7 months; *p* = 0.0056), or CK17^–^CD10^–^ (3 months vs. 10 months; *p* = 0.0015) (Figure 3B–D).

Altogether, our results showed that the tumor expression patterns of MUC6, CK17, CD10, and villin constitute candidate markers with potential prognostic value for patients with primary adenocarcinoma of the gallbladder. According to receiver-operating-characteristic (ROC) curve analysis, tumor expression patterns MUC6^–^CK17^+^ (for patients with well- to moderately differentiated tumors) and CK17^+^CD10^–^ (for patients with poorly differentiated tumors) may be clinically useful, given that the area under the curve (AUC) and *p* values were >0.75 and <0.01, as suggested in the report of Fan et al. [22], respectively (Figure 3E,F).

## 4. Discussion and Conclusions

GBC is a digestive neoplasm characterized by its late diagnosis, high incidence, and mortality in some geographical regions, particularly in the female Chilean population [1,2,3]. Probably due to the relatively low incidences in industrialized Western countries, this disease has not been sufficiently characterized [2]. Currently, there are no reliable tumor-associated protein markers that can be precisely associated with primary gallbladder adenocarcinoma. To the extent of our knowledge, this is the first report evaluating the expression and association of an extensive number of gastrointestinal tumor-associated markers with patient prognosis in a large cohort of primary GBC patients. The obtained results may be of clinical utility, although further validation in an extended patient cohort is needed.

In our cohort, the majority of patients were women (77.8%), in line with evidence reported in the literature, as GBC is two to six times more common in women than in men [2,3]. Estrogen increases the saturation of cholesterol in the bile, thus increasing the risk of gallstone formation, which is believed to be the primary mechanism behind the greater risk of GBC in the female population [23]. In this study, OS was better in women than in men, particularly in younger (early onset) patients with well- or moderately differentiated tumors. These gender- and age-related associations with GBC prognosis were previously reported, both in patients receiving and not receiving postoperative therapies [24,25,26,27]. To date, the mechanistic causes explaining why gallbladder and biliary diseases (including GBC) are more common in women but more severe in men are not clearly known, but a similar behavior was also observed in other female-hormone-driving cancers, such as breast cancer [28], suggesting a dual role of estrogens in increasing the frequency but decreasing the severity of these kinds of tumors.

Of the 17 analyzed gastrointestinal tumor-associated markers, only four (MUC6, CK17, CD10, and Villin) were found to be significantly associated with OS of GBC patients. Mucins are high-molecular-weight glycoproteins that play an important role in protecting the gastrointestinal tract epithelium. However, during disease progression, the expression profile of mucins in cancer cells is altered, and some mucins show a correlation with good or bad prognosis, depending on malignancies [29,30]. Indeed, mucins can protect tumor cells from host immune responses [29], whereas others can also inhibit tumors from spreading and confine tumor cells to the primary site [30]. Cancer antigen-125 (CA125), also known as MUC16, is a large membrane glycoprotein belonging to the mucin family. Following its discovery in the blood of some patients with specific types of cancers, CA125 was used as a tumor marker of ovarian cancer [31]. In addition, the serum levels of CA125 were proposed as an independent risk factor with a diagnosis value for GBC [32]. Higher levels of serum CA125 were associated with a worse prognosis in GBC patients [33]. Although we found that the frequency of CA125 expression was significantly increased in GBC patients with advanced-stage (34.8%) compared to early stage tumors (0%), we did not find a statistical association between CA125 expression and the survival time of patients (data not shown). These data suggest that CA125 serum levels are more sensitive as a predictive prognosis biomarker for GBC than CA125 expression in tumor tissue.

MUC6 is a pyloric-gland-type-secreted mucin that was reported to be involved in tumor progression and metastasis [29,30,34]. MUC6 can retard the invasiveness of breast cancer cells by forming a barrier to prevent tumor cells from spreading [30]. There are several reports showing that MUC6 expression decreases in advanced stages of cancer, particularly in cholangiocarcinoma (CCA), a type of biliary-tree cancer, and its expression was positively correlated with increased patient survival time [35,36,37]. In these studies, it was suggested that the downregulation of MUC6 might be necessary for CCA spreading. To the extent of our knowledge, our report is the first showing a positive correlation between MUC6 expression in tumor tissue and a better prognosis for GBC patients. As in the reports for CCA, our data also indicate MUC6 expression loss in advanced stages of GBC (38.7% of advanced tumors were positive compared to 64.7% for early stage tumors). MUC6 expression was also only correlated with better prognosis in women patients with well- to moderately differentiated tumors, and this correlation is lost in men or women with poorly differentiated tumors. This agrees with the invasiveness-preventing role suggested for MUC6 [30], but once the tumor progresses, its protective role could be overcome by the dedifferentiation of the tumor cells. There are no data in the literature showing a gender-specific tumor protective role for tumor-tissue MUC6 expression, but our data suggest that this point needs to be addressed.

Cytokeratin 17 (CK17) is a basal/myoepithelial cell keratin that is normally expressed in the respiratory epithelium, urothelium, and various glands. In tumors, CK17 is mainly expressed in squamous, basal, and urothelial carcinomas, and in adenocarcinomas, with squamous differentiation [38]. CK17 expression was associated with cancer invasiveness and poor prognosis in stomach, breast, and ovary cancers, and as our results showed, also in GBC [39,40,41]. In addition, our data showed that CK17 was associated with poor prognosis only in patients with poorly differentiated tumors (and exclusive in the female group), but not in patients with more differentiated adenocarcinomas. Men with well- to moderately differentiated tumors also showed an inverse correlation between CK17 tumor expression and OS time, although the low number of included patients limits the statistical significance of this observation. These gender- and tumor-differentiation-status-specific associations between CK17 and OS have not been previously reported in GBC or other cancer patients.

CD10 is a cell-surface enzyme with neutral metalloendopeptidase activity, and it has been used as a marker for intestinal epithelial brush border. It is also expressed in normal bile ducts and gallbladder epithelia but is absent in CCA. Although its expression has not yet been associated with GBC prognosis, the high expression of CD10 was demonstrated to be associated with aggressive tumor behavior and poor prognosis in several types of cancer patients, including prostate [42], melanoma [43], and esophageal squamous cell carcinoma [44]. Interestingly, we observed that GBC patients with tumors co-expressing CD10 and CK17 had a worse prognosis than patients with the opposite profile, suggesting that CD10 and CK17 could be promising co-targets for GBC treatment.

Taken together, our results also allow us to suggest a panel of markers for the detection of an immunohistochemical profile (CK7^+^CK20^–^CK19^+^CKHMW^+^CKLMW^+^MUC1^+^MUC2^–^Vimentin^–^), present in 70.6% of analyzed samples, which could be useful for GBC diagnosis and may particularly help in the identification of the tissue origin of adenocarcinomas of the unknown primary site [45]. However, additional research is needed in order to determine the sensitivity and specificity of this panel for GBC diagnosis. A limitation of our study is the lack of an independent-validation patient cohort, which is a challenging task to be approached in the future, given the rare nature of GBC. Cancer protein biomarkers are increasingly used in clinical practice for diagnosis and prognosis, the identification of responsive patients, and the prediction of treatment outcome [46]. Modern medicine in cancer treatment requires the identification of biological markers that guide in the decision making of such therapy to the chosen patients for more effective results. The rapid breakthrough and development of microarray and sequencing technologies constitute the main tools for paving the way toward individualized biomarker-driven cancer therapy [46].

## Figures and Tables

**Figure 1 diagnostics-11-00153-f001:**
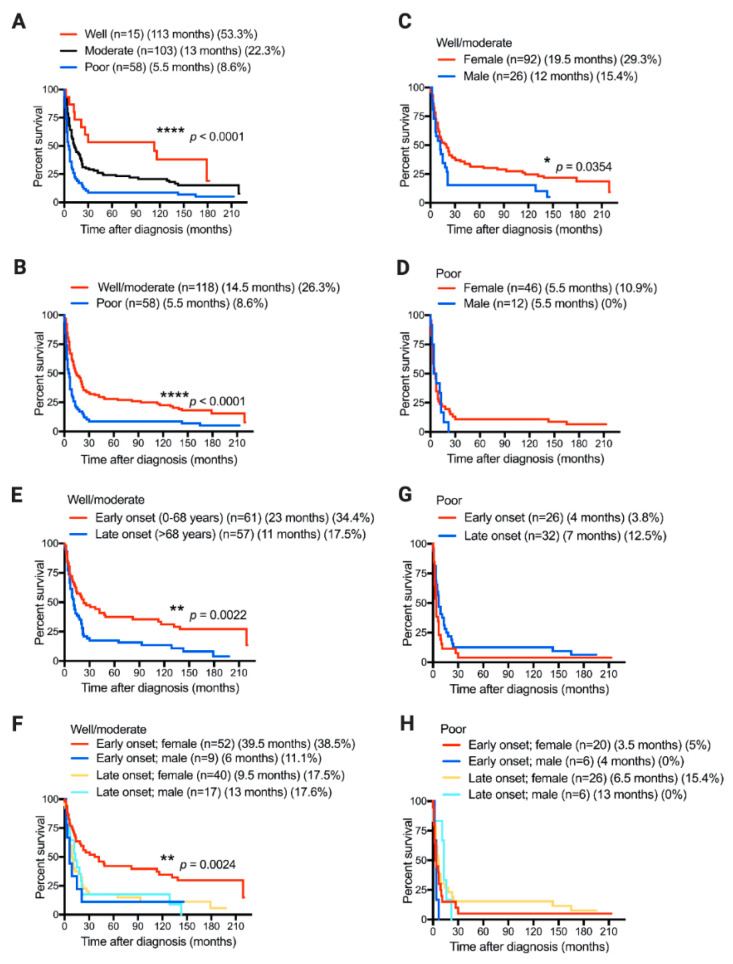
Tumor differentiation status, gender, and age of disease onset correlate with overall survival of gallbladder-cancer (GBC) patients. Kaplan–Meier postdiagnosis overall survival (OS) estimation of GBC patients according to (**A**,**B**) tumor differentiation status, (**C**,**D**) gender, (**E**,**G**) age of disease onset, or (**F**,**H**) age of disease onset and gender together. Each graph shows the number of patients included in each group (*n*), median OS time in months, and five-year survival rate (%). Only *p* values < 0.05 shown.

**Figure 2 diagnostics-11-00153-f002:**
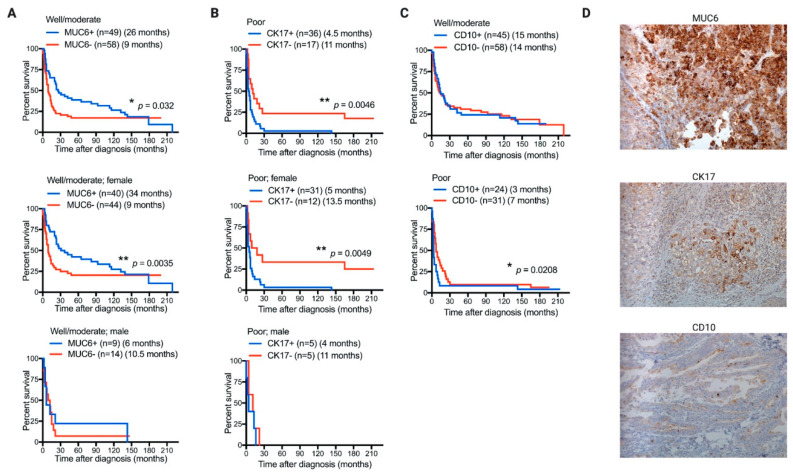
MUC6, CK17, and CD10 tumor expression correlate with the overall survival of GBC patients. Kaplan–Meier postdiagnosis overall-survival (OS) estimation of GBC patients according to tumor expression pattern for MUC6 ((**A**) upper, patients with well- to moderately differentiated tumors; (**A**) middle, female patients with well- to moderately differentiated tumors; and (**A**) bottom, male patients with well- to moderately differentiated tumors); CK17 ((**B**) upper, patients with poorly differentiated tumors; (**B**) middle, female patients with poorly differentiated tumors; and (**B**) bottom, male patients with poorly differentiated tumors); and CD10 ((**C**) upper, patients with well- to moderately differentiated tumors and (**C**) bottom, patients with poorly differentiated tumors). Each graph shows the number of patients included in each group (*n*) and median OS time in months. Only *p* values < 0.05 shown. (**D**) Immunohistochemical representative photographs using a tissue microarray (magnification, X20), for (upper) MUC6, (middle) CK17, and (bottom) CD10.

**Figure 3 diagnostics-11-00153-f003:**
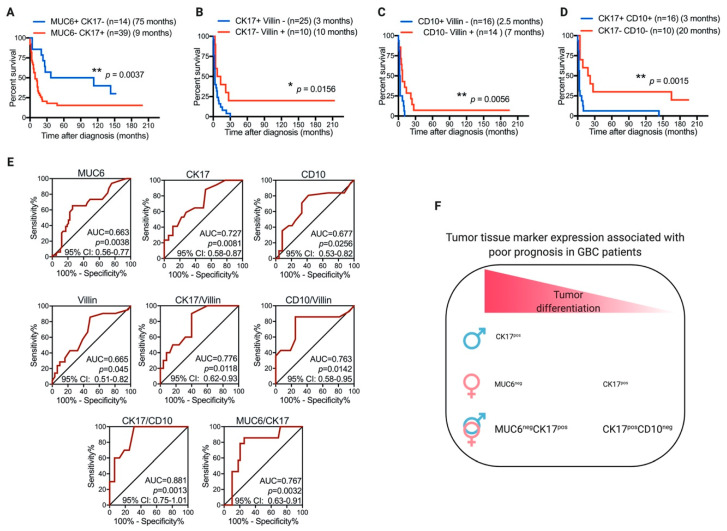
Tumor expression patterns MUC6^+^CK17^–^ and CK17^+^CD10^–^ associated with better and worse prognoses of GBC patients, respectively. Kaplan–Meier postdiagnosis overall-survival (OS) estimation of GBC patients according to tumor expression pattern for (**A**) MUC6/CK17 in well- to moderately differentiated tumors, (**B**) CK17/Villin, (**C**) CD10/Villin, and (**D**) CK17/CD10 in poorly differentiated tumors. Each graph shows the number of patients included in each group (*n*) and median OS time in months. (**E**) Receiver-operating-characteristic (ROC) curves for patient survival according to tumor-marker pattern expressions. AUC: area under the curve. CI: confidence interval. (**F**) Association of tumor-marker expression and poor prognosis of GBC according to patient tumor differentiation status and gender. Font size indicative of statistical significance.

**Table 1 diagnostics-11-00153-t001:** Clinicopathological characteristics of gallbladder cancer (GBC) patients.

CHARACTERISTICS		
Gender	Female	77.8% (*n* = 140)
	Male	22.2% (*n* = 40)
Median age (SD)	All	67.5 (±12.3)
	Female	66.5 (±12.6)
	Male	70 (±11.5)
Tumor differentiation	Poor	32.8% (59 cases) 79.7% female; 20.3% male Median age: 67 (±13.1) Tumor staging: Early (TIS–T1): 0% Advanced (T2–T4): 100%
	Moderate	58.3% (105 cases) 75.2% female; 24.8% male Median age: 69 (±11.5) Tumor staging: Early (TIS–T1): 13% Advanced (T2–T4): 87%
	Well	8.9% (16 cases) 87.5% female; 12.5% male Median age: 60 (±14.8) Tumor staging: Early (TIS–T1): 45% Advanced (T2–T4): 55%

SD: standard deviation; TIS: tumor in situ.

**Table 2 diagnostics-11-00153-t002:** Association between tumor marker expression and tumor differentiation status or tumor staging in GBC patients.

Marker	Total Positive	Tumor Differentiation, # Cases + (%)			Tumor Staging, # Cases + (%)		
		Good/Moderate	Poor	Statistics ^a^	Early ^b^	Advanced ^c^	Statistics ^a^
CK7	164 (97.6%)	112 (99.1%)	52 (94.5%)	NS	16 (100%)	131 (98.5%)	NS
CK20	26 (15.4%)	21 (18.3%)	5 (9.3%)	NS	5 (29.4%)	20 (15.1%)	NS
CK17	118 (72.8%)	81 (74.3%)	37 (69.8%)	NS	8 (53.3%)	97 (75.8%)	NS
CK19	166 (98.2%)	112 (97.4%)	54 (100%)	NS	16 (94.1%)	132 (98.5%)	NS
CKLMW	169 (100%)	114 (100%)	54 (100%)	NS	16 (100%)	132 (100%)	NS
CKHMW	153 (91.6%)	104 (92%)	49 (90.7%)	NS	13 (92.9%)	123 (92.5%)	NS
VIMENTIN	7 (3.9%)	1 (0.8%)	6 (10.7%)	*p* = 0.0014	0	7 (5%)	NS
MUC1	165 (97.1%)	113 (97.4%)	52 (96.3%)	NS	15 (100%)	130 (97%)	NS
MUC2	5 (3.1%)	2 (1.9%)	3 (5.8%)	NS	0	4 (3.2%)	NS
MUC5AC	135 (81.8%)	93 (83%)	42 (79.2%)	NS	15 (88.2%)	107 (82.3%)	NS
MUC6	68 (41%)	52 (45.6%)	16 (30.8%)	NS	11 (64.7%)	50 (38.7%)	*p* = 0.04
CDX2	120 (72.7%)	86 (76.8%)	34 (64.1%)	NS	12 (85.7%)	92 (70.8%)	NS
CEA	99 (57.9%)	64 (55.2%)	35 (63.6%)	NS	8 (47.1%)	82 (61.2%)	NS
CA125	51 (30%)	31 (26%)	20 (39.2%)	NS	0	46 (34.8%)	*p* = 0.009
CD10	73 (44.2%)	50 (45.4%)	23 (41.8%)	NS	10 (62.5%)	53 (40.4%)	NS
CLAUDIN4	132 (81%)	90 (80.4%)	42 (80.8%)	NS	12 (85.7%)	106 (81.5%)	NS
VILLIN	81 (46.8%)	62 (51.2%)	19 (36.5%)	NS	11 (61.1%)	61 (45.9%)	NS

NS: not significant; ^a^ chi-squared test (Yale correction); ^b^ including tumor in situ (TIS) and T1 stages; ^c^ including from T2 to T4 stages.

**Table 3 diagnostics-11-00153-t003:** Univariate and multivariate Cox survival analysis of GBC patients.

Patients with poorly differentiated tumors (*n* = 58).											
**Variable**		**Univariate Analysis**					**Multivariate Analysis**				
		**HR**	**95% CI**		***p* value**		**HR**	**95% CI**		***p* value**	
MUC1		1.37	0.54–3.52		0.509		0.76	0.05–10.89		0.843	
MUC2		1.48	0.98–2.23		0.061		1.78	0.83–3.83		0.138	
MUC5AC		0.81	0.48–1.36		0.421		0.74	0.29–1.89		0.532	
MUC6		0.9	0.6–1.36		0.631		0.62	0.3–1.31		0.212	
CK7		1.66	0.64–4.3		0.292		0.68	0.11–4.23		0.678	
CK20		1.29	0.77–2.16		0.338		1.27	0.53–3.03		0.586	
CK17		2	1.26–3.19		**0.003**		2.46	1.06–5.74		**0.037**	
CK19		1.76	0.62–4.94		0.285		0.63	0.01–38.41		0.824	
CKHMW		1.7	0.8–3.59		0.166		2.87	0.96–8.54		0.058	
CKLMW		1.85	0.56–6.03		0.31		0.1	0.005–1.85		0.121	
CDX2		0.88	0.52–1.47		0.616		0.55	0.24–1.28		0.17	
CD10		1.7	1.08–2.69		**0.023**		3.76	1.65–8.56		**0.002**	
CEA		1.12	0.67–1.89		0.664		1.03	0.45–2.35		0.95	
CA125		1.52	1.02–2.25		0.038		1.99	0.98–4.07		0.058	
Vimentin		1.49	0.94–2.36		0.09		0.74	0.32–1.75		0.5	
Villin		0.79	0.52–1.22		0.289		0.95	0.45–2.03		0.903	
Claudin 4		1.21	0.63–2.32		0.571		0.96	0.4–2.34		0.931	
Age		0.75	0.44–1.28		0.289		1.71	0.82–3.57		0.152	
Gender		1.16	0.6–2.24		0.649		0.97	0.42–2.26		0.954	
Patients with well- to moderately differentiated tumors (*n* = 118).											
**Variable**	**Univariate Analysis**					**Multivariate Analysis**					
	**HR**		**95% CI**	***p* Value**		**HR**			**95% CI**		***p* Value**
MUC1	0.68		0.38–1.21	0.194		0.91			0.38–2.14		0.825
MUC2	0.8		0.59–1.09	0.154		0.96			0.61–1.53		0.878
MUC5AC	0.87		0.58–1.3	0.507		1.13			0.66–1.92		0.665
MUC6	0.69		0.5–0.96	**0.029**		0.46			0.24–0.86		**0.015**
CK7	0.57		0.27–1.2	0.143		0.71			0.46–1.28		0.138
CK20	0.8		0.57–1.11	0.182		0.89			0.58–1.37		0.614
CK17	1.01		0.72–1.4	0.966		1.21			0.78–1.87		0.384
CK19	1.08		0.59–1.98	0.792		1.48			0.66–3.35		0.342
CKHMW	0.74		0.48–1.14	0.17		0.82			0.43–1.56		0.543
CKLMW	1.08		0.56–2.08	0.823		1.72			0.67–4.42		0.259
CDX2	0.72		0.5–1.01	0.06		0.6			0.39–0.93		0.023
CD10	0.95		0.71–1.26	0.706		1.23			0.82–1.84		0.309
CEA	1.03		0.74–1.44	0.846		0.96			0.63–1.46		0.859
CA125	1.18		0.84–1.65	0.335		1.35			0.86–2–1		0.188
Vimentin	1		0.53–1.88	0.996		1.32			0.56–3.11		0.524
Villin	0.86		0.62–1.21	0.396		0.85			0.55–1.32		0.48
Claudin 4	0.77		0.54–1.11	0.169		0.7			0.42–1.14		0.152
Age	1.86		1.24–2.81	**0.003**		2.28			1.43–3.64		**0.001**
Gender	1.63		1.02–2.6	0.041		1.16			0.67–2.02		0.598

HR: hazard ratio; CI: confidence interval. HR calculated as positive versus negative expression for tumor markers or late-age versus early age onset for age variable.

## Data Availability

The data presented in this study are available on reasonable request from the corresponding authors.

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
