# Peer review of "The Evaluation of 17 Gastrointestinal Tumor Markers Reveals Prognosis Value for MUC6, CK17, and CD10 in Gallbladder-Cancer Patients"

_diagnostics, 2021, doi:10.3390/diagnostics11020153_

Round 1
Reviewer 1 Report
In this manuscript, the authors investigated the potential of 17 gastrointestinal tumor-associated proteins (CK7, CK17, CK19, CK20, CKLMW, CKHMW, MUC1, MUC2, MUC5AC, MUC6, CA125, CD10, CEA, Vimentin, Villin, Claudin4 and CDX2) as prognosis markers for gallbladder cancer (GBC) patients. By performing immunohistochemistry staining of tissue microarray comprising 180 primary gallbladder adenocarcinomas, they demonstrated that detection of MUC6, CK17 and CD10 expressions in tumor can be applied to survival prediction for GBC patients. I have some concerns listed below.
- In addition to multivariate analyses, univariate analyses of risk factors should be carried out. Besides, it is not clear what variables were included in the Cox proportional hazard regression model.
- Figure 3, what does the 5% cutoff mean? What was the state variable used in the ROC curves?
- Line 312, it should be “that” but not “than”.
- Significant editing is required for this manuscript.
Author Response
We really appreciate the time and recommendations of the referee. Next a point-by-point response to the concerns.
- In addition to multivariate analyses, univariate analyses of risk factors should be carried out. Besides, it is not clear what variables were included in the Cox proportional hazard regression model.
Answer: In the new version of Table 3 we also show the univariate analysis of HR and the variables included in the Cox proportional hazard regression model (the expression of the tumor markers, age and gender of patients).
- Figure 3, what does the 5% cutoff mean? What was the state variable used in the ROC curves?
Answer: We apologies for including the terms “5% cutoff” in ROC curves, this was a mistake, and we corrected that. We included the 95% confidence interval (CI) for each ROC curve in the new Figure 3.
We performed ROC analysis using GraphPad Prism 7.0. The state (control) variable used for each ROC was the survival times of patients with the opposite marker expression pattern (i.e, for the MUC6+CK17- pattern the state variable was MUC6-CK17+).
- Line 312, it should be “that” but not “than”.
- Significant editing is required for this manuscript.
Answers: We really appreciate the indication for English editing. We corrected that and other mistakes. Moreover, the revised manuscript version has undergone English language editing by MDPI (certificate enclosed). The text has been checked for correct use of grammar and common technical terms, and edited to a level suitable for reporting research in a scholarly journal. MDPI uses experienced, native English speaking editors.

Reviewer 2 Report
Authors present a simple retrospective analysis of ICH staining of 180 gallbladder cancers. They correlated results with prognosis.
I think the paper does not bring anything novel to the field. I would suggest a higher number of analyses. NGS analysis would be of higher interest. Moreover, blood samples and match between tissue and blood would be of interest as well.
Author Response
Gallbladder cancer is a very rare oncological disease in western countries, so the information related to the expression of proteins and their impact on the survival of patients is very sporadic. In general, reports at the level of gene expression are more common, although cohorts of more than 100 patients are scarce. We agree with the referee on the need to address NGS studies and comparative biological techniques derived from peripheral blood samples (liquid biopsies), which we will try in the future, however we think that the information provided in this work is valuable and allows us to project those future studies.
Round 2
Reviewer 1 Report
The authors have addressed all my concerns.
Author Response
Thank you
Reviewer 2 Report
Please improve your analyses. IHC may be of interest only if a prior NGS panel is performed, at least on some patients. You can consider results of NGS and test results with IHC. In this way, the paper will be much more of interest.
Author Response
Although the reviewer suggestions could be interesting, they are impractical (demanded for major changes in all aspects of the manuscript, which seems a bit unfair), and in no case will guarantee an improved impact of the observations.
Numerous (the large majority) of protein IHC-based tumor markers published articles do not include NGS analysis and are still highly relevant, as exemplified with some selected Diagnostics recent papers:
1) Winther, et al. AZGP1 Protein Expression in Hormone-Naïve Advanced Prostate Cancer Treated with Primary Androgen Deprivation Therapy. Diagnostics 2020, 10, 520.
2) Chiokadze, et al. Beyond Uterine Natural Killer Cell Numbers in Unexplained Recurrent Pregnancy Loss: Combined Analysis of CD45, CD56, CD16, CD57, and CD138. Diagnostics 2020, 10, 650.
3) Chung, et al. Nuclear Expression Loss of SSBP2 Is Associated with Poor Prognostic Factors in Colorectal Adenocarcinoma. Diagnostics 2020, 10, 1097.
We thanks for your kind consideration
Round 3
Reviewer 2 Report
Acceptable for publication